# Hydroinformatics education – The Water Informatics in Science and Engineering (WISE) Centre for Doctoral Training

Thorsten Wagener[1,2,9*], Dragan Savic[3,4], David Butler[4], Reza Ahmadian[5], Tom Arnot[6], Jonathan Dawes[7], Slobodan Djordjevic[4], Roger Falconer[5], Raziyeh Farmani[4], Debbie Ford[4], Jan Hofman[3,6], Zoran Kapelan[4,8], Shunqi Pan[5], Ross Woods[1,2]

[1]Department of Civil Engineering, University of Bristol, UK
[2]Cabot Institute, University of Bristol, UK
[3]KWR Water Research Institute, NL
[4]Centre for Water Systems, College of Engineering, Mathematics and Physical Sciences, University of Exeter, UK
[5]Hydro-environmental Research Centre, School of Engineering, Cardiff University, UK
[6]Water Innovation & Research Centre & Department of Chemical Engineering, University of Bath, UK
[7]Institute for Mathematical Innovation and Department of Mathematical Sciences, University of Bath, UK
[8]Department of Water Management, Delft University of Technology, NL
[9]Institute of Environmental Science and Geography, University of Potsdam, Germany

*Correspondence to*: Thorsten Wagener (thorsten.wagener@bristol.ac.uk)

**Abstract.** The Water Informatics in Science and Engineering Centre for Doctoral Training (WISE CDT) offers a postgraduate programme that fosters enhanced levels of innovation and collaboration by training a cohort of engineers and scientists at the boundary of water informatics, science and engineering. The WISE CDT was established in 2014 with funding from the UK Engineering and Physical Sciences Research Council (EPSRC) amongst the Universities of Bath, Bristol, Cardiff and Exeter. The WISE CDT will ultimately graduate over 80 PhD candidates trained in a non-traditional 4-year UK doctoral programme that integrates teaching and research elements in close collaboration with a range of industrial partners. WISE focuses on cohort-based education and equips the PhD candidates with a wide range of skills developed through workshops and other activities to maximise candidate abilities and experiences. We discuss the need for, the structure and results of the WISE CDT, which has been ongoing from 2013-22 (final year of graduation). We conclude with lessons learned and an outlook for PhD training, based on our experience with this programme.

## 1 Introduction

The global water cycle consists of a complex web of interacting physical, biogeochemical, ecological and human systems (Gleeson et al., 2020). Management of this complex cycle has been practised for decades, but new challenges lie ahead due to climate change, growing population pressure, increasing urbanisation and other human-caused environmental disturbances. These challenges can only be addressed through fundamental changes in how we interact with our environment, both in perspective and in practice. The recent focus on the role of water security in addressing ecosystem services and sustainability has further emphasised the need for new approaches to achieve this dual goal (UNEP, 2009; 2011; 2017). In the UK, the

government's 25-year Environment Plan (2018) clearly signals the importance and value of the environment, including "reducing our carbon emissions and building resilience against the extreme weather associated with climate change."

Infrastructure is considered equally important as evidenced by the UK National Infrastructure Assessment (2018), which regards water and waste infrastructure as being essential for health and wellbeing, environmental sustainability and economic stability, and which points to investment of £44 billion in the water sector. At the European level, the EC sponsored ICT4Water Cluster has published its Digital Single Market for Water Services Action Plan (2018). Supporting these efforts in turn requires new, whole-system, multi-faceted, data-intensive, interdisciplinary approaches to research, training and innovation –

approaches which take advantage of the information explosion and leading-edge technologies of the 21st century (Blöschl et al., 2012; Ceola et al., 2015; Habib et al., 2012; Jonker et al., 2012; King et al., 2012; Montanari et al., 2013; Ruddell and Wagener, 2013; Seibert et al., 2013; Thompson et al., 2012; Wagener et al., 2010; 2012).

As the capabilities of digital devices soar and their prices plummet, sensors are providing greater amounts of information than ever, at lower costs and with greater reliability than previously possible (Mao et al., 2018). Opportunities for real-time

monitoring and management are increasing dramatically; so is access to far more powerful Information and Communication Technology (ICT) tools and devices (ICT4Water Cluster, 2018). These tools enable 'People as sensors' (crowd-sourcing, citizen science), bringing together the skills of humans to observe and interpret with the interconnection of the Internet to enable new types of information to be crowd-sourced (Seibert et al., 2019). Combining these trends provides opportunities to address both old and new problems in innovative ways to meet emerging challenges around the water cycle. Globally, it is

estimated that savings of $7.1bn to 12.5bn per annum (SENSUS, 2012) may be realised through the adoption of smart water technologies to minimize operational inefficiencies and to maximize the effectiveness of capital and operational expenditure. Management of the water cycle, a system characterised by inherent complexity, heterogeneity, and uncertainty (not least because of linked social, natural and engineered systems), has already gained from advances in computing, ICT and Hydroinformatics resulting in new technologies deployed in engineering practice (Romano et al., 2014). The increasing non-

stationarity of the global climate system, and the subsequent implications for the terrestrial water cycle, will significantly change how we approach the problem of long-term planning and of estimating hydrologic design variables (Brown et al., 2015; Milly et al., 2008; Sivapalan and Blöschl, 2015). So, while new data become available as mentioned above, historical data will lose some of its value for long-term analysis, e.g., if catchments have undergone significant land use change, experience more extreme rainfalls than previously recorded or see significantly changed streamflow characteristics due to the building of human

infrastructure (Jain and Lall, 2001). Long-term planning in non-stationary systems requires a move away from traditional empirical approaches (widely used in operational engineering hydrology), towards process-based models which are necessarily more complex and require deeper process understanding (Milly et al., 2008; Wagener et al., 2010; Clark et al., 2015; Musolini et al., 2020). Process-based models – increasingly with unprecedented resolutions – will also demand better computational skills of their users to utilize them effectively and thus influence training and education (Hut et al., 2017).

In its 2030 vision document "The Value of Water", Water Europe, as the recognised stakeholder platform of the European water sector, promotes a future-proof European model for a water-smart society that requires a paradigm shift in how water is managed (Water Europe, 2020). Bluefield Research assessed that the smart water sector in Europe and the USA is increasing rapidly, estimating that by 2025 it will be worth $11 billion and $12 billion respectively. A total of 61% of companies offering digital water solutions in the US market have been founded since the year 2000 (Smart Water Magazine, 2020). The need for

this shift, and the tremendous market opportunity it brings, is set against the backdrop of widespread concern for the future of the water sector and its workforce, as for example reported for the UK in a recent employer survey (CIWEM, 2016). The survey found that 81% of employers saw increased staff turnover and 70% said that skills shortages had reduced their capacity to deliver projects. Hence the requirement for hybrid skills and expertise in water and informatics will grow with the need for increasingly intelligent systems. New water industry roles such as 'digital e-service delivery', 'smart water networks' and 'big

data analytics', are misaligned with established single-discipline postgraduate programmes, and require new 'hybrid' skill sets not normally taught as a package.

For society to take full advantage of leading-edge technologies we need to provide training for hydroinformaticians, i.e. scientists and engineers capable of working at the interface of traditionally separate disciplines of informatics, science and engineering, to manage information and water cycles effectively (Fig. 1) (Popescu et al., 2012; Merwade and Ruddell, 2012;

Makropoulos and Savic, 2019). Reports by the Council for Science and Technology (CST, 2009), the UK Royal Academy of Engineering (RAE, 2012) and the UK Institution of Civil Engineers (ICE, 2012) have highlighted a particular shortage of engineers and scientists in industries of national importance for the UK, such as "energy, water, sanitation, communications and IT systems". In response to the projected skills shortage in the IT sector in Europe, the European Commission has launched a 'grand coalition' to tackle this shortage (CEDEFOP, 2018). It is difficult to see how the need for skilled engineers working

at the interface of IT with water science and engineering can be met by IT graduates alone – who would also have to be educated in water processes subsequent to their IT training. We rather need to train scientists and engineers who can work at the interface of traditionally separate informatics, science and engineering disciplines to fill these 'hybrid job' roles.

The Water Informatics in Science and Engineering Centre for Doctoral Training (WISE CDT, http://wisecdt.org.uk/) aims to fill the skills gap discussed above by offering a postgraduate programme that fosters enhanced levels of innovation and

collaboration to train a cohort of engineers and scientists at the boundary of water informatics, science and engineering. Disciplinary breadth, a focus on PhD candidate cohort experience, and collaborative effort (not least, between four research-intensive UK universities in delivering the programme) are its key novel features. The WISE CDT was established in 2014 with funding from the UK Engineering and Physical Sciences Research Council (EPSRC) and led by the University of Exeter in partnership with the universities of Bath, Bristol and Cardiff. The WISE CDT inducted its first cohort of PhD candidates

in October 2014 and thus far it has recruited 84 candidates and graduated 26 (as of April 2021). The final group of candidates was recruited in 2018 and is expected to graduate in 2022 (Fig. 2). The initial Principal Investigator was Professor Dragan

Savic (2014-2018) and now it is Professor David Butler. Here we describe the programme, show some selected educational elements and discuss what the WISE CDT has achieved so far.

## 2 The WISE CDT Training Approach

There is growing evidence that producing highly skilled researchers and future leaders requires doctoral supervisory teams to support: (a) candidates specialist disciplinary development, and (b) their wider skills development (Roberts, 2002; Buckley et al., 2009; Brodin and Avery, 2020). Such a 'T-shaped competency profile' of broad general and in-depth disciplinary skills has been identified as crucial for future water professionals (Uhlenbrook and de Jong, 2012). The WISE CDT programme addresses both sets of skills through:

(i) candidates participating in a two-semester WISE CDT Postgraduate School involving both broader disciplinary and wider research methodology training, thereby increasing candidates' research skills and knowledge base, while enhancing their exposure to interdisciplinary work in ways suitable for careers within or outside academia;

(ii) further participation in specialist Master's level modules that candidates can attend at WISE CDT partner institutions, after the WISE CDT Postgraduate School;

(iii) a strategy of 'guided freedom' that provides the candidates with opportunities to be active partners in shaping their learning experiences;

(iv) formalised career development and placement experience at overseas academic or industry partners; and

(v) further transferable and leadership skills development that enhances candidates' career and project management skills, ensuring they make a successful transition to their career of choice.

**2.1 WISE CDT Postgraduate School in Water Management and Informatics**

Each WISE CDT PhD lasts four years (Table 1): the first two semesters (October to May) are allocated to a WISE CDT Postgraduate School in Water Management and Informatics (run at the University of Exeter, and including lectures from staff at Cardiff, Bath and Bristol Universities as well as our industrial and international partners). The residential School ensures that a functioning cohort of WISE CDT PhD candidates is immediately established, thereby reducing the risk of attrition due

to the feeling of not belonging to an academic community (McAlpine et al., 2009) or due to social isolation (Ali and Kohun, 2007). The structured approach to the WISE CDT cohort/community development applies for the duration of the programme involving the four-stage framework for dealing with social isolation developed by Ali and Kohun (2007). The WISE CDT framework addresses: (i) pre-admission to enrolment (orientation, administrative liaison, formal social and induction events), (ii) first year (integration, cohort approach, ice-breaking, buddy system, research proposal development, supervisor selection);

(iii) second year through to thesis writing (collaborative model, topic presentation and feedback, Summer School, transferable / leadership skills), and (iv) thesis stage (structure for the thesis, collaborative model, face-to-face communication).

As the programme is aimed at a diverse set of graduates from Engineering, Environmental Science, Geographical Science, Physics, Mathematics, and Computer Science; candidates devote the first two semesters to taking an appropriate set of existing postgraduate level modules at Exeter to cover their knowledge gaps as well as to advance their skills in water processes and informatics. The cohort's basic programming skill is brought to a similar level (with some candidates advancing clearly above this level) by a "learning by doing" model with a software development project related to one of the water cycle themes (Fig. 1). The project and module offerings are augmented by specialist ICT modules, by staff from partner institutions (both national and international) and industry, including modules on, for example, software development, cloud computing, object-oriented programming, cyberinfrastructure, etc. This training includes a three-day intensive training course, which has been arranged for each of the cohorts early in the WISE CDT Postgraduate School, to provide researchers with practical experience of software development techniques. The IT focused teaching components are complemented by "water knowledge" components in which each partner institution trains candidates in their specific area of strength (Exeter: urban drainage, Bath: wastewater treatment, Bristol: hydrology, Cardiff: hydraulics).

This programme of training ensures that all candidates gain a solid understanding of water informatics, of knowledge across water topics and of wider research methodology before they develop and commence on their PhD research project. The Postgraduate School programme comprises eight taught Master's level modules, worth a total of 120 credits. These include:

- Hydroinformatics Tools,
- Urban Drainage and Wastewater Management,
- Water Supply and Distribution Management,
- Environmental and Computational Hydraulics,
- Computational Hydrology,
- Mathematical Modelling of Wastewater Treatment Processes,
- Programming for Engineering,
- Research Methodology.

WISE CDT candidates across the cohorts appreciated the opportunity to attend a first year of courses, which is uncommon for a PhD programme in the UK, where traditional PhD's are typically based on individual research only. We selected some candidate quotes from our regular surveys (discussed further below) to highlight this aspect.

Candidate James Webber (University of Exeter, Cohort 1): "I have found that, as my PhD research has developed, the modules from the Postgraduate School have become very useful. A good example of this is computer programming and coding, which now forms a substantial part of my research, despite me knowing very little about it before starting the programme."

Olivia Bailey (University of Bath, Cohort 2): "The postgraduate school in Exeter helped to broaden my knowledge of the water world and gave me confidence to get out of my comfort zone and develop my PhD work in a direction that straddles multiple engineering disciplines".

Stephanie Mueller (Cardiff University, Cohort 4): "I am currently in my first year of the WISE CDT programme, attending the Postgraduate School at the University of Exeter. This year has allowed me to receive an overview of the different research areas of the four partner universities, as well as giving me the opportunity to explore new ideas, methodologies and inspirations which have helped shape my research project. In addition to the scientific knowledge that I have gained, I have also become an integral member of my cohort group. Coming from a different country meant that I faced many challenges regarding language, culture, regulations etc., but the support of my fellow peers, better known as my "WISE-Family", has helped me to overcome these challenges, for which I am very grateful. Ultimately, the WISE CDT has been a great opportunity for me to undertake my PhD adventure in the UK."

Georgios Sarailidis (University of Bristol, Cohort 5): "Unlike other doctoral programmes, the WISE CDT offers the opportunity to take part in a wide range of courses and activities such as transferable skills modules and research seminars, which have helped me to gain valuable skills, explore new ideas and methodologies and meet new people from both industry and other academic institutions. Finally, the cohort-based structure of this programme was undoubtedly one of the most appealing points. Gathering people with different academic backgrounds has provided us with an opportunity to support each other and develop a comradeship throughout our first year. In particular, as I come from a different country I have had to deal with many new things and my peers have provided me with lots of support, helping me to make this transition much easier."

Further familiarisation with facilities and potential supervisors from the other three universities during the Postgraduate School is ensured by seminar visits arranged at each of the partner institutions. Candidates pursue further specialist skills training available at partner institutions after the Postgraduate School. These taught components in Years 2-4 (selected based on candidate background and intended specialisation) are chosen from postgraduate level modules offered by the partner institutions in agreement with supervisors.

## 2.2 Transferable Skills and Leadership Programme

The need for improvement in the development of research careers and training in transferable skills was highlighted in Sir Gareth Roberts' report (Roberts, 2002), which led to new funding for generic skills training and further calls for the development of research skills at both the PhD and postdoctoral career stages in the UK (RCUK, 2010). The WISE CDT offers extensive and structured provision of transferable and leadership skills through the Transferable Skills and Leadership Programme arranged during the WISE CDT Postgraduate School and over the subsequent years for each cohort in multi-day blocks, as listed in Table 1. In Year 1 (in Exeter), the programme concentrates on the acquisition of a deeper understanding of the research process and methodology, together with project management for researchers, personal effectiveness, communication skills (both written and oral), relationship between science and society, and introduction to ethics. In subsequent years all candidates attend residential programmes which cover a wide range of areas including: Year 2 (in Bristol) – preparation & delivery of conference talks and posters (including an external consultant teaching a one-day workshop titled 'with confidence at conferences'), and writing & refereeing of journal articles; Year 3 (in Cardiff) – management & team

working skills; professional etiquette, planning & writing a thesis, thinking of one's career, and entrepreneurship & leadership skills; Year 4 (in Bath) – viva preparation, strategic problem formulation, knowledge exchange & research exploitation, and early career development.

## 2.3 WISE CDT Summer School

An annual week-long residential Summer School (incorporating a mini conference) provides an opportunity for supervisors and PhD candidates from all cohorts and diverse disciplinary backgrounds to share their experiences of working on water informatics related projects while networking with other staff and candidates (Fig. 3). They are joined by the Strategic Advisory Board, which includes representatives from key stakeholders (e.g. water companies, Government agencies, and consultancies), for part of the week. The Summer School serves candidates from Years 1 to 4, with emphasis on encouraging discussion and

exchange of ideas across disciplinary boundaries (i.e., natural sciences, engineering, humanities and social sciences). Each Summer School is organised around a central water informatics challenge, including: [1] 'Water Hackathons' – an intensive competition of brainstorming and computer programming that draws together the talent and creativity of participants – who were tasked to identify water sector challenges and to develop mobile phone app ideas to address these; [2] Developing a proposal to improve our global understanding of a relevant water issues for submission to a fictional international agency; [3]

A water game – where candidates produced a water-themed board game. The winning group created an educational game, named 'Hydropolis', which focused on water management challenges faced by developing cities; [4] Water security challenge – where candidates were asked to develop a business plan for a UK water utility which focussed on addressing long-term water security; [5] Design of Flood Defences in Devon – where candidates worked with a local authority and the Environment Agency to propose innovative flood defence approaches in the Torbay area (UK) in the context of climate change, and [6] A

currently planned event – a specialist in the design of infographics has been hired to run a workshop for the PhD candidates, to be followed by group projects where candidate teams are asked to design water infographics.

Workshops, mini group projects and networking sessions during the Summer School have been organised, not only to address the main Summer School topic, but also to help in developing both specific and transferable research skills, and generic skills, to prepare participants for future careers both inside and outside of academia. The participants meet every morning with

215 internationally recognized academics (e.g. Elena Toth, University of Bologna; Patrick Reed, Cornell University) and leading industrial experts (including those from non-technical disciplines, e.g., business, ethics, law, psychology, marketing) to discuss their ideas and receive feedback. The afternoon is reserved for development activities. The Summer School's main activity finishes with participants pitching their ideas to a 'Dragon's Den' style panel of academic and industry representatives that judge the outcomes and award prizes to the best ideas. Dragon's Den is a British TV Series where entrepreneurs attempt to

220 sell their business ideas. During the Summer School, a one-day mini conference/symposium is organised to allow participants to present their research ideas and progress on their PhD projects. These activities again involve outside agencies to help both candidates and academics embed creative problem-solving approaches within the CDT. The Summer School is also used for

a PhD progress monitoring meeting (performed by the Program Management Group - PMG), where progression to the next year of the research programme depends on satisfactory performance in the previous year. An annual written report and a supervisor and mentor meeting report are used to assess the candidate's progress fairly and impartially, as well as to give candidates the opportunity to raise any concerns.

## 2.4 Supervisory Arrangements - Guided Freedom Strategy

As the primary responsibility for the major aspects of PhD training rest jointly with the candidate and the supervisors we proposed a strategy of guided freedom (following ideas by Prof Willem Bouten, University of Amsterdam, based on Personal Communication with Prof Jasper Vrugt, UC Irvine) to facilitate the process of empowering the candidates and supervisory teams to create a successful doctoral training experience. We implemented the strategy in two stages, each involving a number of activities.

The first stage, which takes place during the Postgraduate School (Year 1 of the PhD programme), involves the following activities: (i) a bi-weekly candidate-led journal paper review seminar, (ii) the first Summer School (including candidate project discussions), and (iii) research proposal development. Paper review seminars initially involve academic members of staff in guiding paper selection (across the topics covered by the WISE CDT programme) and critical review, but with responsibilities gradually passed onto the candidate cohort itself. An activity, in which the first-year candidates discuss, develop, and choose three research topic areas in agreement with potential academic supervisors and partners, runs throughout the year. Candidate choices are then reviewed, one topical area is selected, and candidate-supervisor connections are established, with the research proposal being presented at the Summer School event. During the first stage, the candidates also participate in weekly cohort seminars/meetings linked to the Postgraduate School. The first stage of the programme finishes with research proposal presentations and allocation of the supervisory teams. The time between the Summer School and the beginning of Year 2 is focused on refining the research proposal and planning.

Each supervisory team involves a primary project supervisor and a second supervisor at the institution where candidates are registered. The supervisory team also includes a mentor who can make an objective assessment of the candidate's progress, provide pastoral support, and monitor the all-important working relationship between the (primary) supervisor and candidate. We encourage that, in addition to the supervisory team at the primary institution, candidates also have a supervisor at one of the other institutions, who provides a complementary skillset or expertise. In addition to enhancing the candidate's training, this interaction also promotes cross-institutional research. The supervisory team is completed by an industrial advisor in cases where the project is co-funded by a sponsoring organisation. We also encourage supervisory teams where early career academic colleagues are working alongside experienced PhD supervisors (the early career academic is normally the lead supervisor). During the second stage of their PhD, CDT candidates have regular meetings with the supervisory team at the institution where they are registered (commonly every 1-2 weeks), while also participating in the regular research group meetings (organised by the well-established institutional research groups).

## 2.5 WISE CDT Management Structure

Management of the WISE CDT is structured as follows. Day to day management of the WISE CDT is provided by the lead Principal Investigator (PI) and the WISE Postgraduate School Manager at the University of Exeter (the latter is an academic staff member with 50%-time availability), with support of a full-time administrative position. Each partner university has a Co-Director (Co-D) and a Co-Investigator (Co-I) as local management team (10%-time commitment), supported by a part-time administrator (20%-time commitment). We found that two academic staff members per university (Co-D and Co-I) are necessary so that regular meeting participation and continuous candidate support can be assured. The PI, Co-Ds and Co-Is make up the Program Management Group (PMG), which meets once every 3 months to review progress, consult with candidates, and make CDT management decisions. A Strategic Advisory Board meets annually to receive reports on all aspects of the CDT operation, and provide external input and recommendations.

## 3 Candidate Experience

A key aspect of any CDT is to develop and embed ways of working that enhance the research candidate experience with the goal of graduating well-balanced and better prepared PhD candidates. To strengthen the feeling of belonging to the WISE CDT academic community and to reduce social isolation, the WISE CDT programme established peer-support groups (a 'buddy scheme'), whereby all incoming first year candidates are assigned to a second, third or final year PhD candidate at Exeter who has volunteered to become a 'buddy'. This effort is followed by a similar system being established for them when they move to the university where they are registered for the remainder of the programme. All first year PhDs are invited to meet their buddies at the beginning of the WISE CDT Postgraduate School. There is no pressure to do so, but candidates are regularly made aware that there is someone who they can contact to ask for advice.

### 3.1 Candidate Participation and Feedback

We regularly gather feedback from our candidates, through surveys, individual feedback or via the cohort representatives. The "Surveys" would cover the following,
- End of Year 1: Postgraduate School review meeting – face-to-face feedback to Centre Manager and program Director;
- Transferable Skills and Leadership module evaluation forms;
- Annual Progress Review "Happiness Index";
- End of programme candidate Experience Survey.

Everything is reported to the PMG (directors and co-Is) and discussed at their quarterly meetings. Data from the Happiness Index and Candidate Experience Surveys are also reported to the external Strategic Advisory Board. In addition, we informally expect candidate representatives to "survey" their cohort to feed in ideas, comments and criticism to each quarterly PMG meeting.

Each cohort of PhD candidates elects a representative during their first year on the programme. Representatives actively contribute to PMG meetings to feed in comments from their peers, to participate in discussions and to make recommendations and bring in ideas. They are present for all Open Business agenda items (i.e. only excluding Closed Business items where CDT financial and individual candidate matters are discussed), receive minutes of all meetings and contribute agenda items for discussion. Each cohort maintains a closed social media group to share feedback and provide peer support. Every year, the WISE CDT uses the candidate feedback to trigger actions and various modifications of the programme.

In response to a request from the CDT Strategic Advisory Board, a "happiness index" question was incorporated into candidates' 2018-19 Annual Progress Review forms. This question asked candidates to rate their general happiness in their PhD on a scale from 1-5 (from "very unhappy" to "very happy"). 63 current candidates across cohorts 2-5 answered this question, with 70% overall assessing themselves to be either "happy" or "very happy". There appeared to be no correlation between happiness rating and progress, although generally each cohort was "happier" than the previous one. We considered this could relate to the ongoing improvements being made to the programme but might also indicate the increasing pressures felt when nearing completion of a PhD. The CDT will continue to use the happiness index question in future years, enabling valuable longitudinal studies. The experience gained through running the CDT has provided insight as to the best scheduling of events or inclusion of new content or activities. Comprehensive candidate feedback data complements this by identifying the points at which declining satisfaction is more likely, enabling proactive interventions to be made. We also share the CDT's candidate experience data (in anonymized form) and other metrics amongst the partner organisations so that good practice is disseminated.

A candidate experience survey was undertaken for our first cohort on completion of their PhD program. This survey asked candidates a range of questions about their experience of the CDT, aiming to find out what had been valuable, what could be improved, and what difference the WISE CDT had made to them. The survey questions incorporated ratings from 1-5 (from "very poor" to "excellent") plus free text comment fields. While the sample size was small, everyone completed and returned the feedback form. Candidates rated the CDT experience overall as "good", with a mean score of 4.25 out of 5. Most frequently mentioned as the best elements were the cohort experience and support and friendships gained, the opportunity for a funded research visit, and the opportunity throughout the programme to present work and engage with other researchers. These results are pleasing, as they represent the areas not generally available on a standard PhD programme. The most frequently cited areas for improvement were: (1) Re-think the postgraduate school - We went from 6 compulsory and 2 optional modules to 8 compulsory modules which reduced candidate options but was required due to University wide changes. We also went from long and thin modules – over a full semester, to short and thick modules – over a few weeks. We further tried to help candidates who came into the program with less quantitative skills to catch up before the actual program started. (2) A unified approach across the four universities (e.g. registration periods, PhD thesis submission, extension requests) – This was difficult to adjust given that these administrative processes were largely out of our control, though communication was improved. (3) More interaction between the four universities, including both candidates and academics (e.g. joint supervision, inter-disciplinary

events, data/software sharing) – This interaction was expanded as WISE grew. This exit survey will be undertaken annually as each cohort completes their programme.

Throughout the CDT, candidate feedback has contributed to the evolution of the CDT structure. The main actions undertaken in response to constructive criticism from candidates are the following:

- Enhanced candidate support / administrative support.
- Obtaining Chartered Institution of Water and Environmental Management (CIWEM) accreditation to meet the needs of candidates without a formal engineering background (they found that this will help their employability with
engineering companies).
- Amendments to content and scheduling of the taught components of the Postgraduate School.
- Enhancements to transferable skills modules, e.g. viva preparation, careers guidance.
- Broadening the Industry Day focus / range of guests to cover the breadth of candidates' research interests.
- Website enhancements – including a secure library of CDT templates / guidance.
- Ongoing engagement with alumni, including in CDT events, e.g. talks to current candidates.
- Involving candidates actively in planning of CDT events.

**3.2 Industry Engagement and Professional Accreditation**

The WISE CDT aims to develop PhD graduates who may progress to academic, industry, regulatory, practitioner or research institutions. In this respect exposure to real industry challenges and projects, and the networking and career development
opportunities that arise from engagement with industry, are highly valuable and necessary components of the training programme delivered by the CDT. To deliver this aspect there are various components of the programme with an industry focus: (i) a series of seminars and invited lectures from industry and water-related stakeholders during the Postgraduate School in Exeter, (ii) an annual WISE CDT Industry Day where the candidates present their project proposals / results to-date to water industry and practitioner stakeholders via a poster and networking session, (iii) engagement with our Strategic Advisory Board
(SAB) via a poster competition (judged by the SAB) held during the annual Summer School, (iv) engagement with professional organisations such as the Chartered Institution of Water & Environmental Management (CIWEM), the Institute of Water (IoW), British Hydrological Society (BHS), UK Water Industry Research (UKWIR), International Association for Hydro-Environment Engineering and Research (IAHR), and the International Water Association (IWA) as relevant. Additionally, many of the PhD projects within the WISE CDT programme are co-developed by the candidates and their supervisor teams
with industry and/or practitioner partners, ensuring a route to impact and adding real-world relevance to the project when it is delivered. Industrial partners have found this level of interaction highly beneficial, e.g. Dan Green, Head of Sustainability and Innovation at Wessex Water said: "Wessex Water is seeing the emergence of skills and knowledge gaps in common with other companies in the water sector …. The WISE CDT helps address these gaps by providing leading interdisciplinary training across the subject areas of water engineering and informatics which are vital for the future of this industry and highly pertinent

to our company". Industry interest has led to jointly funded projects operating within the WISE CDT, with direct cash funding from industry partners in excess of £0.7m, and in-kind contributions such as collaborator staff time and advice, invited seminars, site visits, access to stakeholder data, opportunities for in-company or on-site trials, access to facilities and infrastructure etc., valued in excess of £2m.

Industry engagement with the WISE CDT includes the annual WISE CDT Industry Day, for which the number of UK water
industry companies and stakeholders attending has risen from around 20 to about 40 during the last four instalments (Fig. 4). For our most recent event in February 2020, run in collaboration with the Wet Networks events which are jointly convened by Arup and WRc, we attracted about 100 attendees for the half day programme, with roughly 50% being external guests. Amongst these were water utilities (Bristol Water, Dwr Cymru / Welsh Water, Wessex Water), consultants (Arup, Fraser Nash Consultancy, PA Consulting, RSKW, SWECO, WRc), contractors (Jacobs, Mott Macdonald, MWH Treatment, Stantec,
Wood, WSP), NGOs (Oxfam, RedR), supply chain companies (Craley, DeepRoot, Flow3D, Innovyze, Tre-Altamira), stakeholder organisations (Alliance for Water Stewardship, Water Industry Forum, UK Water Partnership), and UK Government Agencies (Environment Agency, NERC).

WISE also achieved accreditation from the UK Chartered Institution of Water and Environmental Management (CIWEM) programme in June 2018 (www.ciwem.org). We pursued accreditation with CIWEM in direct response to candidate feedback
as we considered that CIWEM accreditation would meet the needs of candidates without an engineering background. This was in fact CIWEM's first accreditation of a PhD course and covers all five cohorts of candidates. Areas of good practice highlighted by the Accreditation Panel included the relationships with and between candidate cohorts, our industry and practitioner links, and the ability of candidates to draw on academic expertise and facilities across the four universities. Achieving this formal accreditation was important for our candidates because following a CIWEM-accredited programme
enhances candidates' career prospects by facilitating their path to Chartered Engineer status in the UK.

Candidate Laura Wignall (University of Exeter, Cohort 3): "As a student member of CIWEM it is great to see the WISE CDT programme acknowledged as a CIWEM accredited course. It demonstrates to future employers that CIWEM recognises the course content as relevant to the professional disciplines in the water and environmental management sector, where many of us aspire to end up working."

Candidate James Webber (University of Exeter, Cohort 1): "The WISE CDT is well structured to teach the fundamental scientific and practical principles of water engineering design to researchers from multi-disciplinary backgrounds. The course is a great gateway to bring new perspectives into environmental engineering – in particular by advancing best practice through linking new computational tools to water management challenges. CIWEM's accreditation of the programme is an excellent benchmark of quality to communicate the benefit of the CDT to future employers and collaborators."

### 3.3 International Research Visits

All WISE CDT candidates are provided with financial support for a 3-month research/industry visit in the UK or abroad (Fig. 5). Most of them spent three months with an international partner institution, about a third visited industrial partners while others went to academic institutions, and some visited both. Table 2 shows the diversity of institutions that our candidates visited.

Olivia Bailey (University of Bath, Cohort 2) visited TU Delft in the Netherlands: "Visiting TU Delft has been the cherry on top of my PhD cake, I feel really proud of what I have achieved. The knowledge and data gained through this visit has really helped to advance the robustness of the last three years of work. Of course, it has also been a great opportunity to improve my cycling proficiency!"

Maria Xenochristou (University of Exeter, Cohort 2) visited the National University of Singapore: "This experience was unique not only in terms of academic enrichment and collaboration but also experiencing the rich and diverse culture of South East Asia."

Stephen Clee (Cardiff University, Cohort 2) visited Hohai University in Nanjing, China: "The experience of being able to visit an international university is highly beneficial both practically and personally – you get to develop your skills and knowledge, work with international academics who are experts in your field and experience a whole new culture at the same time."

### 3.4 Outreach: The Land of the Summer People Art Project

WISE CDT PhD candidates have been involved in a wide range of outreach activities – largely at their home universities after the candidates completed their first year in Exeter. Most outreach activities related to the candidates' research topic, which they only fully engage with in years 2-4 of their PhD. Example outreach activities include: (a) 'Walking with Scientists' – Ioanna Stamataki (Bath Cohort 1) led a guided walking tour showcasing Bath's rich science history – as part of 'FUTURES 2019'. Ioanna's talk focused on the historical floods of the River Avon and the applications of using historical data. (b) 'Tomorrow's Engineering Week' 2019. As part of this initiative, Cardiff candidate Santi Lopez (Cohort 5) volunteered on behalf of ICE Wales to provide 'Engineering Team Challenges' to secondary school students, with the aim of encouraging them to consider a career in engineering. And (c) 'Tomorrow': Swindon's Science Festival. WISE candidates showcased an Augmented Reality Sandbox, developed by the company KeckCAVES and supported by the National Science Foundation in the USA, which allowed the audience to sculpt miniature sand landscapes and generate 'clouds' and 'rainfall' with their hands. The group also demonstrated the effects of flooding (such as flash flooding from a dam break) and natural disasters (e.g. tsunamis) on different landscapes and their associated engineering mitigation strategies. We generally left organizing these activities to the individual PhD candidates and their supervisors/institutions, rather than organizing them centrally with an overarching objective in mind. However, one outreach activity was started centrally, and we believe is worth sharing.

One very worthwhile outreach activity (based on candidate feedback) that took place during the first year of the first PhD cohort was the "The Land of the Summer People" project (https://thelandofthesummerpeople.org/collaborative-process/). This

art-science research collaboration brought together artists – working on environmental issues – and WISE CDT PhD candidates (Fig. 6). The project explored the unstable relationship between society, water and place by exploring flood impacts over the Somerset Levels, a very flood prone region in England. Pairs of one artist and 3 PhD candidates jointly developed art pieces, including paintings and drawings, which were subsequently exhibited in a public gallery in Exeter.

The project started with a series of workshops. First, the PhD candidates came together to gather background knowledge regarding the history and future of flooding over the Somerset Levels. They worked in small groups of three candidates who integrated their findings in a poster so that what they learned was shared with the rest of the group. In a subsequent workshop, candidates and artists were brought together to explain their respective background and interests. Interestingly, it became clear during pre-workshop discussions that both groups were rather nervous about the event. The candidates thought that the artists would consider them to be boring, while the artists assumed that the candidates would think of them as strange. It turned out that both groups actually appreciated the very different viewpoints that the other group presented – challenging their existing mindset. The second workshop included pairing artists and candidates in small groups, which would then go on to develop individual art projects. The small groups self-organized to develop their projects in the outdoors, or in the artists' studios. An interesting challenge for the candidates was to produce outcomes, such as a painting, which, in contrast to a scientific graph, did not have a single specific message, but rather would trigger different emotions in the audience. The candidates have been trained to ensure that 100 viewers of their graphs would walk away with a single message, while the artists wanted them to (preferably) walk away with 100 different emotions and ideas.

The outcomes of this collaboration were ultimately exhibited publicly in a dedicated gallery space in Exeter, UK. Art pieces included paintings and stone masonry, as well as 'flood survival kits' (including sponges, sand, inflatable rubber and information material) which were handed out to the public in Somerset as a starting point for discussion on flooding problems in the area (Fig. 6).

## 4 WISE CDT Candidates' Backgrounds and Follow-on Careers

We have recruited a total of five candidate cohorts into the WISE CDT programme (Fig. 2). Each cohort included between 15 and 18 PhD candidates. In total we recruited 84 candidates, of which 39% were female and 61% male. Candidates' ages on entry ranged from 21 to 50, with 82% of candidates being aged between 20-29 at the start of the programme, 17% between 30-39 and 1% aged 40+. Due to the mix of funding through EPSRC, industry and the four participating universities, we were able to recruit both UK (60%) and EU (40%) candidates into the programme. We used industry and matching University funding to support EU candidates, since EPSRC funding could only be used to support UK candidates. EU candidates originated from Belgium, Denmark, France, Germany, Greece, Italy, Netherlands, Slovenia and Spain. Most candidates entered the programme after finishing a postgraduate Master's degree, while a few had either concluded a four-year undergraduate programme (e.g. MEng) or a Bachelor's degree only. As noted previously, we recruited candidates from a wide range of

science and engineering backgrounds as intended, given our goal to provide a broad-based interdisciplinary research programme (Fig. 7).

A relatively small fraction of the candidates has graduated thus far (mainly from Cohorts 1 and 2) and any assessment of the candidates' subsequent career path is therefore preliminary. Experience in other doctoral programs suggests that strong interactions amongst researchers during the PhD continues to influence their later career paths (Carr et al., 2017; 2018). It will be interesting to see whether such findings are also true for the WISE CDT program. The first group of graduating WISE PhD candidates moved into research positions, consultancy, and other jobs, such as with the regulatory authorities etc. For example,

Josh Myrans (Cohort 1) extended his PhD work to further develop Artificial Intelligence based technology for automated detection of faults in wastewater pipes from CCTV inspections. This technology, developed initially as part of his WISE PhD, is currently being implemented via a knowledge transfer project with a large UK water company. Maria Xenochristou (Cohort 2) is now employed as a postdoc at Stanford University in the USA. She is using the advanced machine learning and other skills gained during her WISE PhD to advance research in the field of bioinformatics. Rosanna Lane (Cohort 2) now works

for the UK Centre for Ecology and Hydrology (UKCEH) in Wallingford, as a hydrological modeller – building on the UK-scale modelling framework she developed and tested, while Mariano Marinari (Cohort 1) is a technical consultant for EcoNomad Solutions Ltd and also teaches Applied Mathematics at a secondary school in his home country of Italy.

**5 Conclusions and Lessons Learned**

Harnessing and exploiting the rapidly growing sources of available data and computational power are among the greatest

professional challenges and opportunities facing water and environmental practitioners as well as researchers today. The proliferation of sensors of various types, large-scale and widespread data acquisition, increasingly sophisticated modelling tools, information and communication technologies, the "Internet of Things", and the roll-out of 5G wireless networks will enable far more 'symbiotic' relationships to be developed between rural populations, city governments, urban citizens and businesses. In the long-term, digital sensors, smart phones and wearable smart devices will together form the primary interface

between customers, other stakeholders and the companies providing water services. Massive growth in the availability of open water data enables a strategy to monitor, understand and simulate our non-stationary water environment in new and exciting ways.

The EPSRC Water Informatics in Science and Engineering (WISE) Centre for Doctoral Training (CDT) is an educational response to these opportunities. We use an educational model in which one-year of initial cohort-based training in one location

is followed by 3 years of subsequent research across all partner institutions. Lina Stein (Cohort 3) comments on the cohort benefits of the WISE CDT: "In the cohort there is a wide array of experiences and problem-solving approaches. This creates an atmosphere of mutual help and a tight-knit group of friends. The close contact will hopefully persist in future years, spanning a research network over four universities". Graduated candidates have moved successfully into industrial, practitioner and academic positions. For example, David Evans, Director of Natural Energy Wyre praised the WISE CDT skills development:

"The WISE CDT provides a unique opportunity for candidates to participate in shaping their own research topic. This, coupled with the wide range of skills they will acquire by completing all that the postgraduate school offers, makes them very attractive prospective recruits for the Water industry". In addition, the WISE CDT places a strong emphasis on building a bridge between academia and the water industry through the program. We initially built on existing links between individual academics and companies but expanded these throughout via dedicated activities such as an annual Industry Day and a very active Strategic

Advisory Board. We believe that the PhD projects (full list on http://wisecdt.org.uk) reflect the ambitious goals of the WISE CDT. Example projects of those candidates who have graduated thus far include "Water demand forecasting using machine learning on weather and smart metering data", "National-scale hydrological modelling of high flows across Great Britain", "Self-powered biosensors for water quality monitoring: sensor development and signal treatment" and "Event Management and Event Response Planning for Smart Water Networks". Several PhD projects were extended as knowledge transfer activities

having secured additional funding.

    Like any comparable large-scale education program (Blöschl et al., 2012; Serlet et al., 2020), we experienced successes and challenges along the way, which we have tried to share in this document. We realized that bringing a very heterogeneous group of PhD candidates together to study requires effort in ensuring that some candidates are not overwhelmed because their background is less quantitative. We also realized that even a seemingly homogeneous PhD training set-up across four UK

universities (as it might look like from outside the UK) reveals significant differences in the approaches taken by individual universities when tested. Solutions to such challenges are transparency and open communication channels for the PhD candidates to be able (and to feel confident) to highlight problems when they occur. Throughout the program, we increased PhD candidate participation in tailoring the program. Overall, the WISE CDT also certainly represented a tremendous opportunity for the academics involved. We created new research partnerships (within and across universities) and we enjoyed

having a large group of excellent junior researchers with the freedom to jointly define research projects with their supervisors without constraints.

    We believe that some of our experiences might be helpful for PhD-level water education more widely, outside of dedicated doctoral programs. In line with the suggestion of one reviewer we list what we believe are some 'must haves' of modern doctoral training based on our experience:

•   We believe that a coordinated baseline training component at the beginning of a PhD is highly beneficial for most candidates. Training in skills like advanced maths, programming, data analytics and similar knowledge is especially useful and best gained through a coordinated training effort, rather than through individual activity. Programs where such training is common, e.g. in the USA, have an advantage over those that traditionally do not offer a training element (e.g. traditional PhD education in the UK).

•   All candidates strongly emphasized that the cohort aspect of our PhD was a major positive during their PhD journey. While it will not always be possible to create cohort-based training, our experience nonetheless suggests that

coordinated efforts to connect the PhD candidates that start within a particular time period will be highly beneficial for those involved.

- Tailored skills training throughout is valuable to help PhD candidates gain the experience needed for writing, presenting, etc. Candidates throughout the course found such training most helpful when it was tailored to their discipline and specific needs, and when it offered wider discussion with experienced academics as part of the training.

The central outcome of the WISE CDT so far is a group of highly trained doctoral candidates and graduates who have been educated in a training environment that was re-designed 'from the ground up', bringing the strengths of four research-intensive universities together for the first time. Our training programme reaches across traditional science and engineering divisions, 515 in line with the skills that future graduates in this area will need. The programme explicitly acknowledges and encourages interdisciplinary collaborations. To provide an enhanced candidate experience we have invested additional time and resources into transferable skills training and built cohorts of candidates – giving them their first professional networks and even life-long friends. This re-designed educational experience is our response to modern industry requirements, for example the urgent need to elevate coding skills in training water practitioners, and to the rapidly expanding opportunities in research. Where 520 exactly the best mix between computing skills and water science and engineering knowledge lies continues to be a topic of ongoing debate (Hut et al., 2017), and no doubt the requirements of training programmes such as the WISE CDT will continue to evolve to meet pressing new challenges, from monitoring of biomarkers for new diseases, to water security, in the future.

**Acknowledgements**

The WISE CDT is funded by the Engineering and Physical Sciences Research Council (EPSRC), grant number EP/L016214/1 525 and by the Universities of Bath, Bristol, Cardiff and Exeter. We gratefully acknowledge additional funding and support from our many academic and industry partners. We thank Alyssa Serlet and the anonymous reviewer for their constructive criticism that helped to improve the paper.

**Author contributions**

TW, DS, DB and DF prepared the draft manuscript. All authors contributed to the final version of the manuscript.

**Competing interests**

The authors declare no competing interests.

## Data availability

There is no data involved in the work presented here beyond that included as tables.

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

**Table 1: WISE CDT programme components.**

|  | Year 1 | Year 2 | Year 3 | Year 4 |
|---|---|---|---|---|
| **Postgraduate School in Water Management and Informatics** | 120 Masters level credits via a combination of long & thin and block modules at Exeter (October to May) |  |  |  |
| **Specialist Masters Level Modules** | Specialist modules based on the agreed research topic and in consultation with supervisor(s) |  |  |  |
| **Transferable and Leadership Skills** | Skills training at Exeter (during the Postgraduate School) | At Bristol (1 week) | At Cardiff (1 week) | At Bath (1 week) |
| **Research project + thesis writing** | From June to September | All year | All year | All year |


**Table 2: Research visit locations.**

| Institution Hosting Research Visit | Host Institution Supervisor |
|---|---|
| Asian Institute of Technology, Bangkok, Thailand | Prof Mukand Babel |
| Centre of Ecology and Hydrology, Wallingford, UK | Dr Cecilia Svensson |
| University of Saskatchewan, Canmore, Canada | Prof Martyn Clark |
| Cornell University, Ithaca, USA | Prof Patrick Reed |
| Delft University of Technology, Delft, Netherlands | Prof Jan Peter van der Hoek |
| Deltares / Delft Technical University, Netherlands | Dr Robert McCall |
| DHI, Hørsholm, Denmark | Dr Ole Mark |
| University of British Columbia, Vancouver, Canada | Dr Aaron Cahill |
| Griffith University, Gold Coast, Australia | Prof Rodger Tomlinson |
| Ludwig-Franzius-Institut, Leibnitz Universität Hanover / Technical University Braunschweig, Hanover, Germany | Dr Stefan Schimmels |
| Hohai University, Nanjing, China | Prof Yongping Chen |
| International Institute for Applied Systems Analysis (IIASA), Laxenburg, Austria | Dr Yoshihide Wada |

| | |
|---|---|
| JBA Consulting | Ms Kirsty Styles |
| KWR Water Research Institute, Utrecht, Netherlands | Dr Mirjam Blokker |
| Kyoto University, Kyoto, Japan | Prof Yasuto Tachikawa |
| Laval University, Quebec, Canada | Prof Sebastien Houde |
| Leibniz University Hannover, Germany | Dr Stefan Schimmels |
| Luxembourg Institute of Science and Technology, Luxembourg | Dr Stan Schymanski |
| Nanjing Normal University/Hohai University, Nanjing, China | Drs Qiang Dai & Jing Huang |
| National University of Science and Technology, Zimbabwe | Dr Eugine Makaya |
| National University of Singapore (NUS), Singapore | Prof Vladan Babovic |
| San Diego State University, USA | Prof Hilary McMillan |
| Singapore Centre for Environmental Science and Engineering (SCELSE), Singapore | Dr Jamie Hinks |
| Hohai University, Nanjing, China | Prof Pei Xin |
| Stellenbosch University, Stellenbosch, South Africa | Dr Wesaal Khan |
| Texas A&M University, College Station, USA | Prof Scott Socolofsky |
| The University of Auckland, Auckland, New Zealand | Dr Heide Friedrich |
| Tsinghua University, Beijing, China | Prof Binliang Lin |
| University College London, UK | Dr Eugeny Buldakov |
| University of Arizona, USA | Prof Tom Meixner |
| University of Bologna, Bologna, Italy | Prof Alberto Montanari |
| University of California, Irvine, California, USA | Prof Brett Sanders |
| University of Canterbury, Christchurch, New Zealand | Prof Roger Nokes |
| University of Melbourne, Melbourne, Australia | Prof Tim Fletcher |
| University of Melbourne, Melbourne, Australia | Dr Murray Peel |
| University of Waterloo, Waterloo, Ontario, Canada | Prof Bryan Tolson |
| University of Zurich (UZH), Zurich, Switzerland | Prof Jan Seibert |
| Washington State Department of Ecology, USA | Dr George Kaminsky |
| WaterHarvest, India | Om Prakash Sharma |
| University of New South Wales, Sydney, Australia | Prof Ian Turner |
| Wuhan University, Wuhan, China | Prof Junqiang Xia |

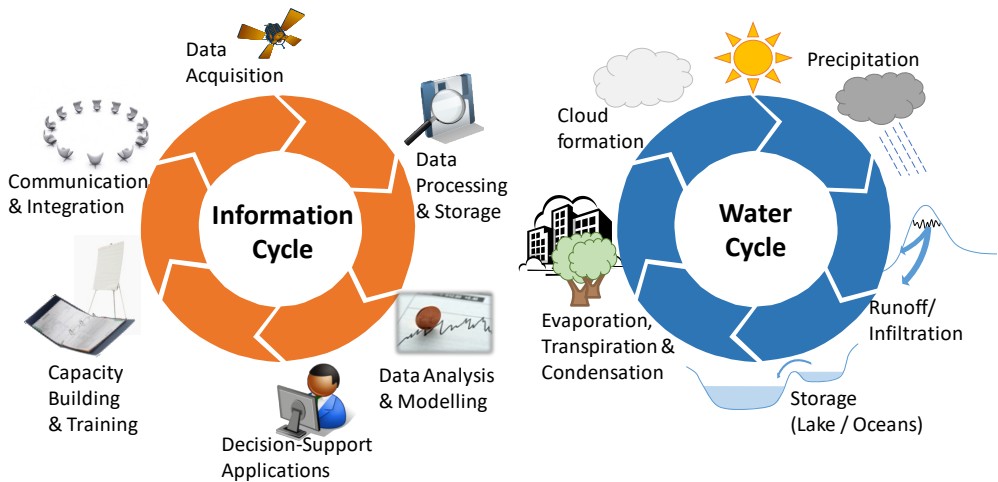

**Figure 1: Information and water cycles.**

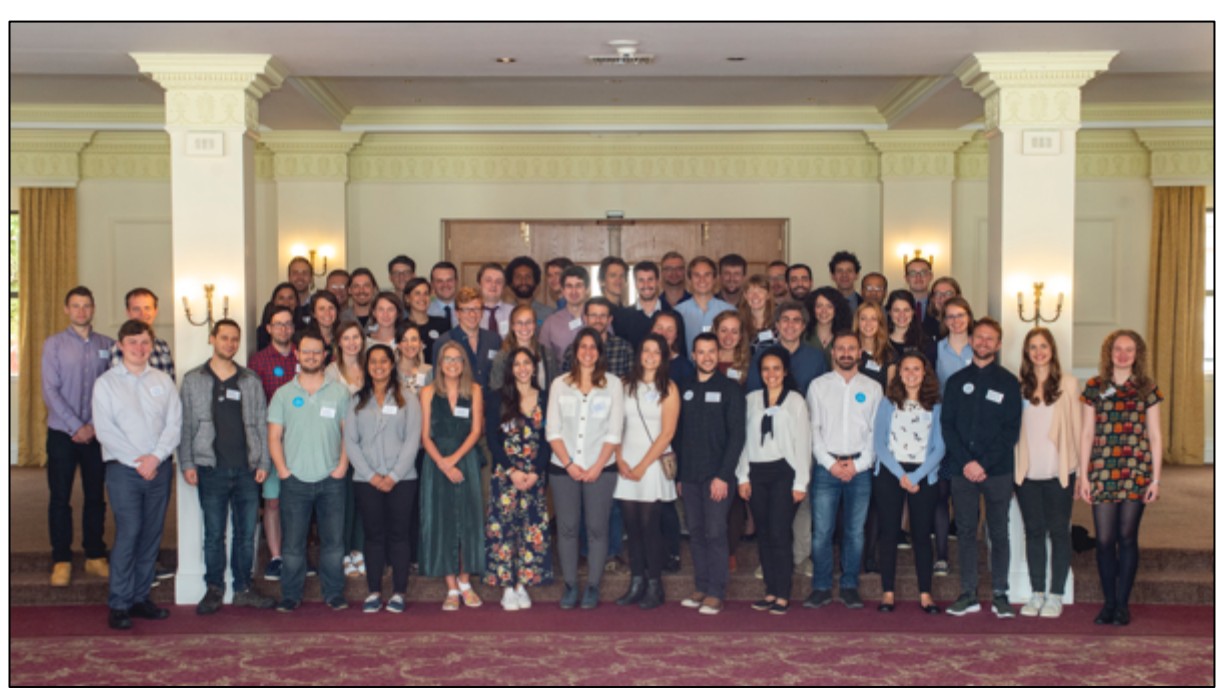

**Figure 2: WISE CDT PhD Students (Summer School in Torquay, UK, 2019, Photo credit © Steven Haywood 2019)**

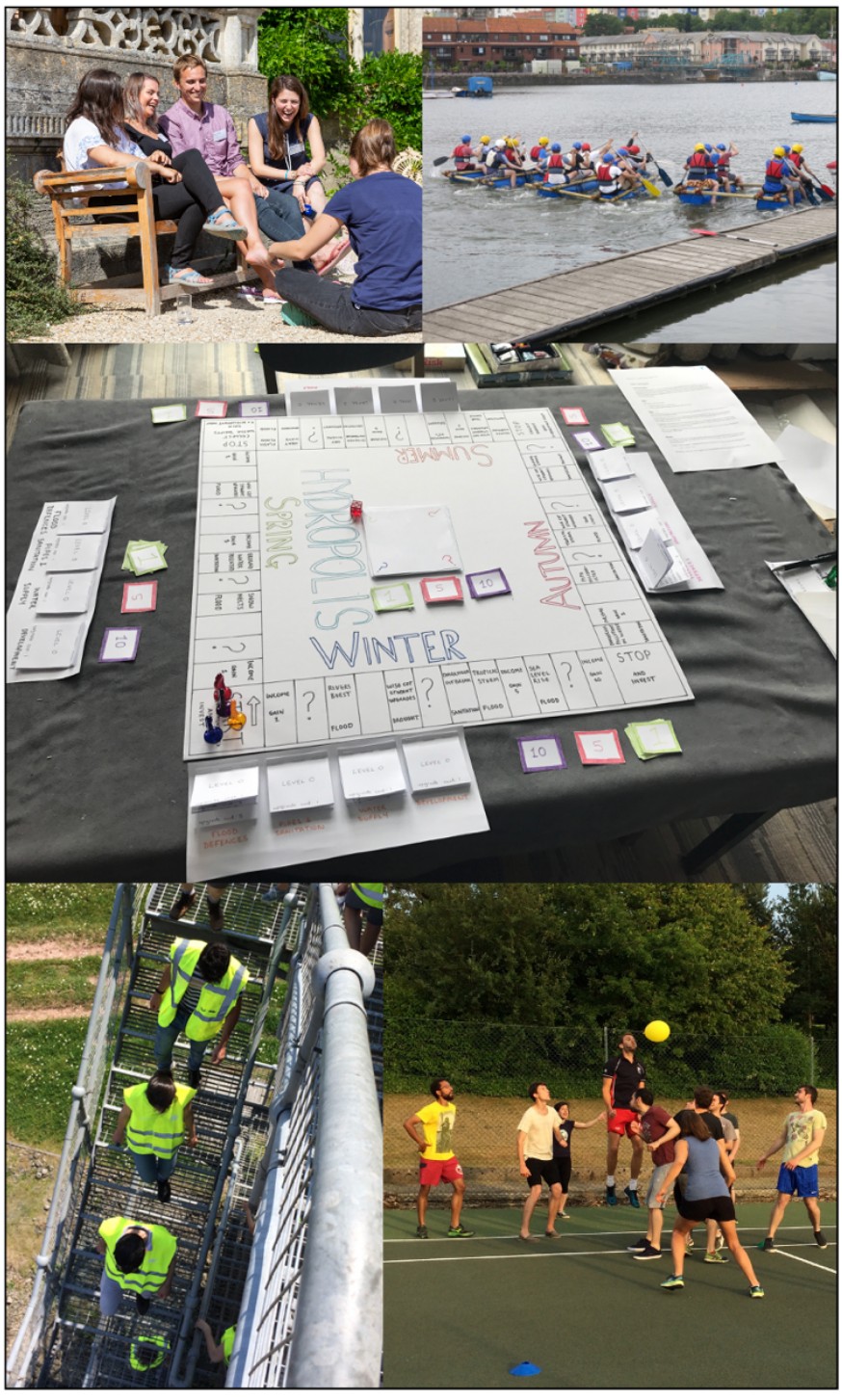

**Figure 3: Examples of Summer School Activities (Photo credits: Top left © Tim Gander 2018. Bottom left and right © Lina Stein 2017).**

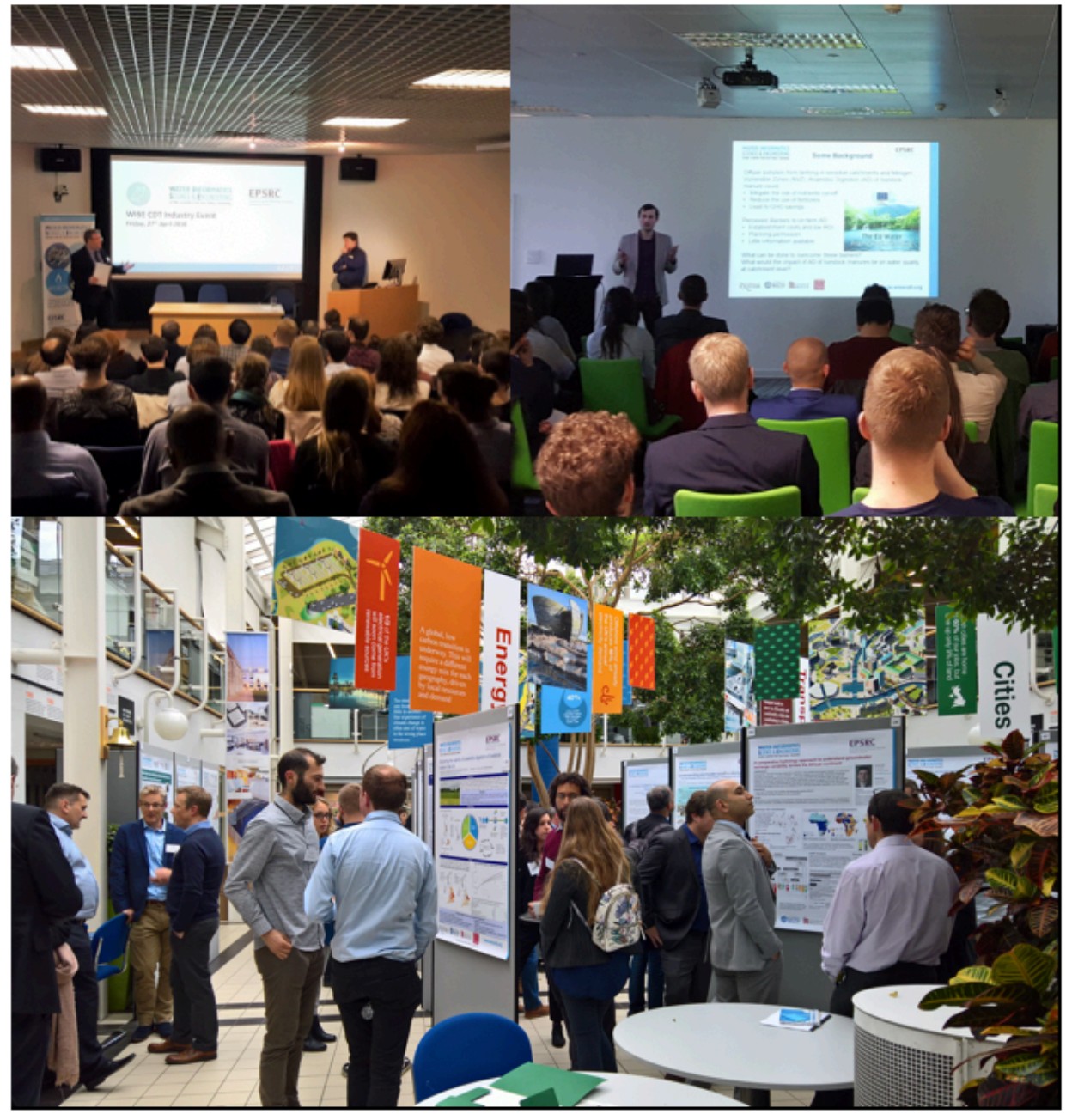

**Figure 4: We have run WISE CDT Industry Days with Atkins in 2016, HR Wallingford in 2017, Arup in 2018, and jointly with the Wet Networks event series in 2020. (Photo credits: © Tom Arnot).**

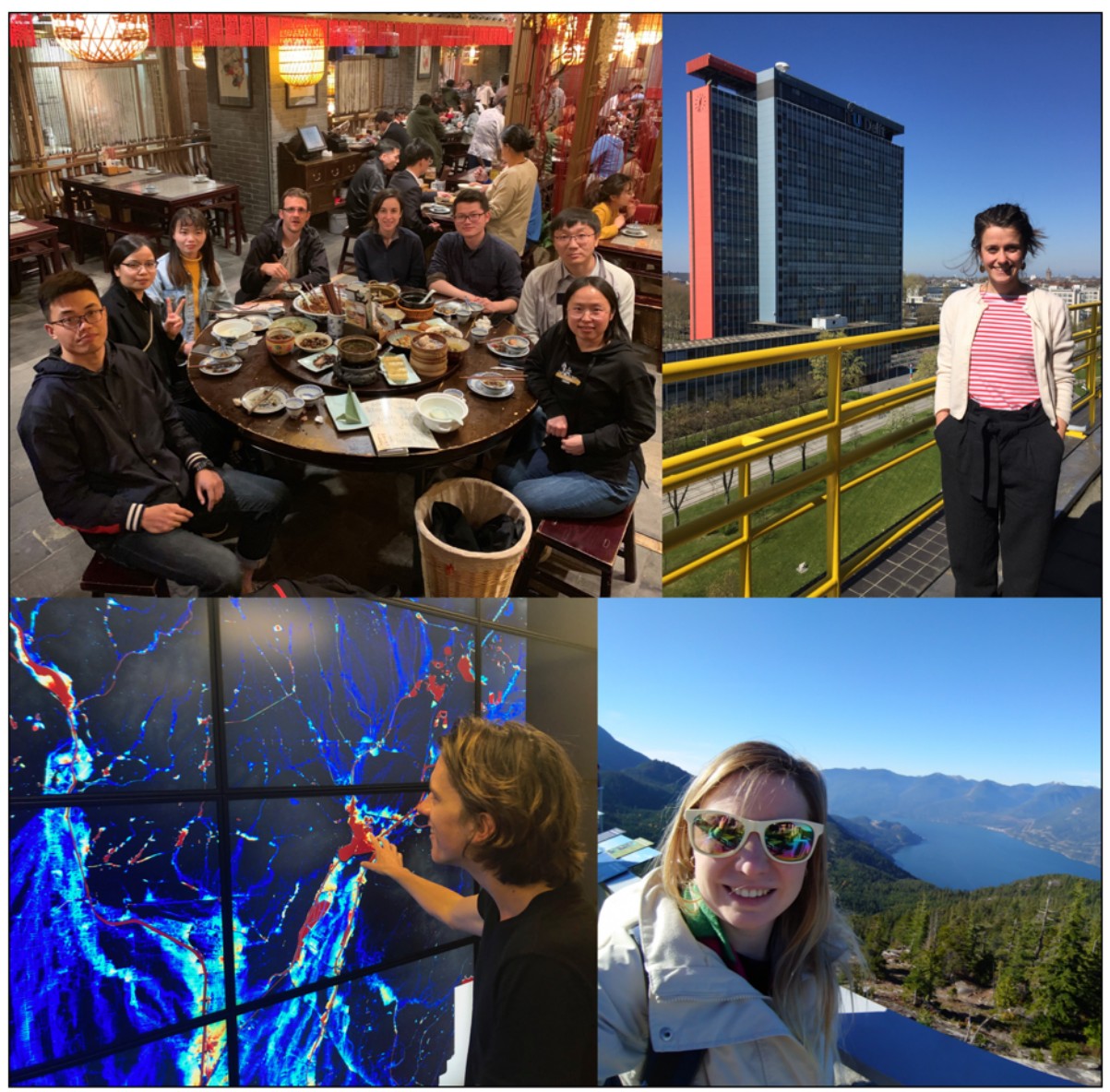

**Figure 5: Clockwise from top left: Anna Lo Jacomo (Cohort 2) and Richard Rees (Cohort 3) enjoying the hospitality of their hosts at Hohai University, Nanjing, China; Cohort 2 student Olivia Bailey's research base at TU Delft, Netherlands; Cohort 2 student Olivia Milton-Thompson making the most of her research visit to UBC, Vancouver, Canada; and Cohort 2 student Joe Shuttleworth at the University of California, Irvine, USA.**


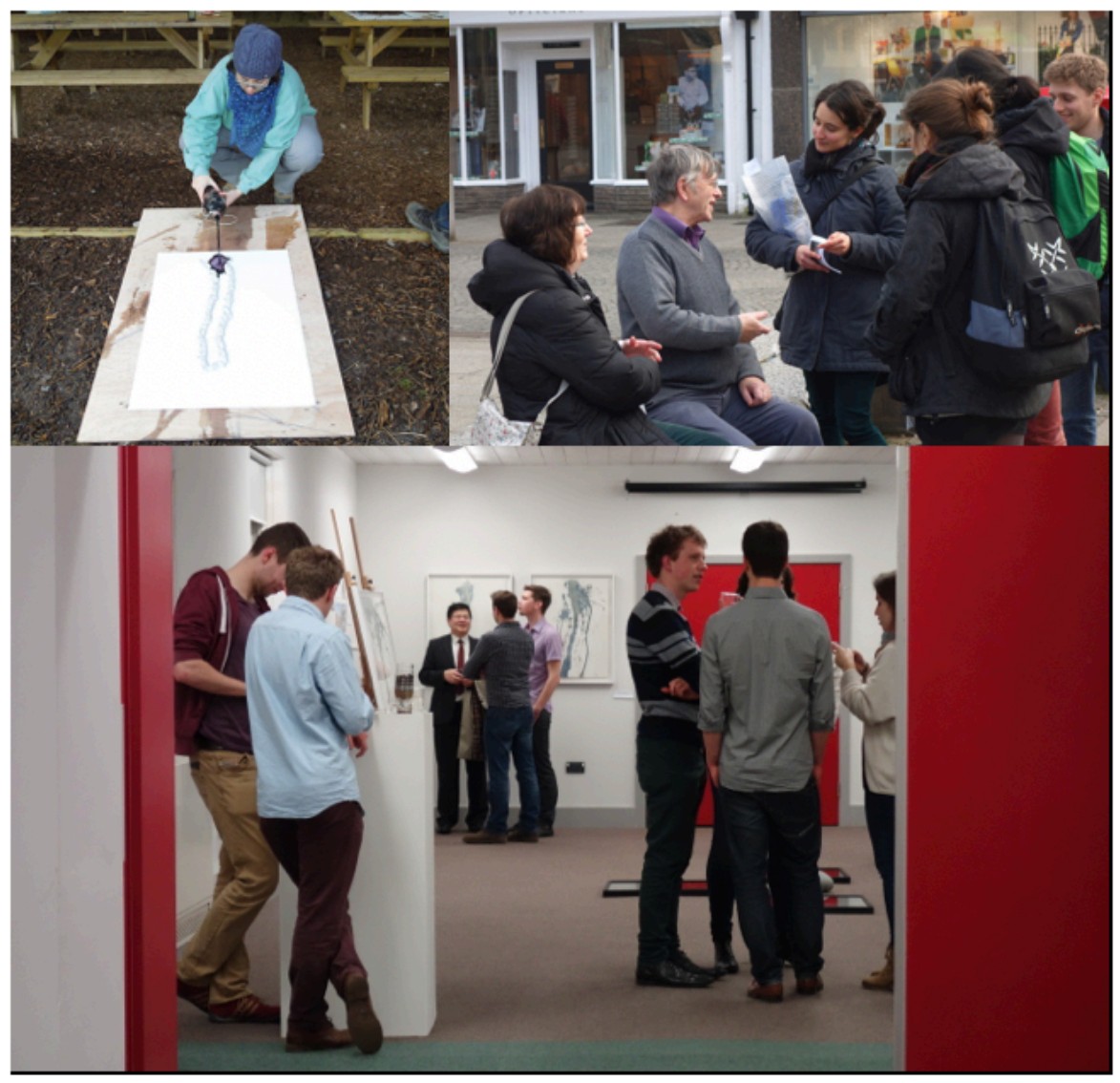

**Figure 6: The Land of the Summer People art project paired each group of 3 PhD students with an artist. Each group independently developed a piece of art. Pieces included paintings, stone masonry as well as 'flood survival kits' to be handed out to the public. (Photo credits: © Seila Fernandez Arconada 2014).**


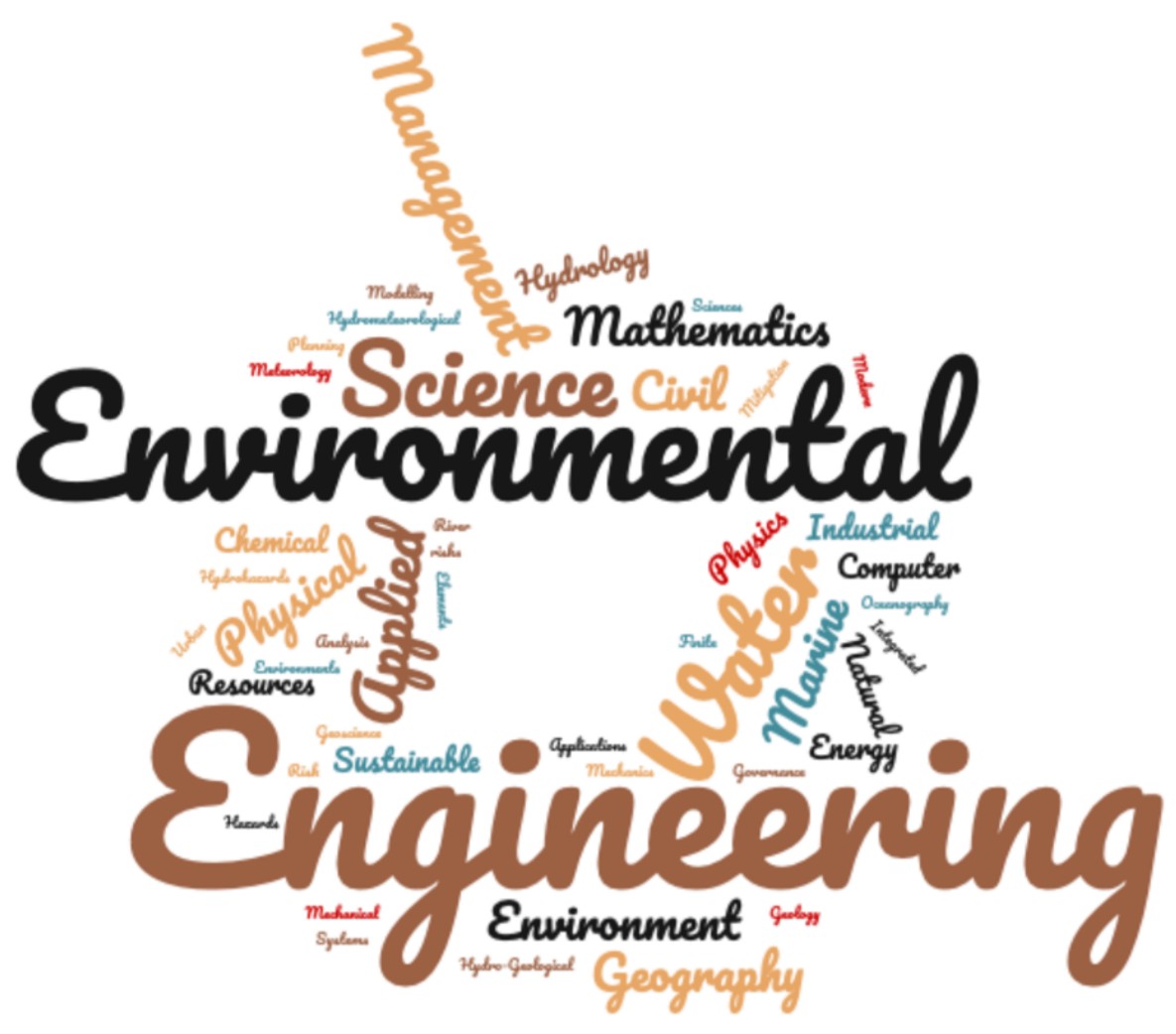

**Figure 7: Word cloud indicating the subject areas in which WISE CDT students had obtained their highest entry qualification before entry to the programme.**