# Peer review of "Hydroinformatics education – The Water Informatics in Science and Engineering (WISE) Centre for Doctoral Training"

_Hydrology and Earth System Sciences, 2020_

## Referee Comment (RC1) · Alyssa Serlet (Referee) · 3 Nov 2020

General comments: I have very much enjoyed reading this paper. I believe doctoral programmes of this nature will be increasing in number in the near future, and papers such as these are of great relevance to the scientific community to learn from the outcomes and experiences of such programmes. The paper is very well written and it explains thoroughly the need and benefits of the presented programme. It is very important to share the success story of this programme; however, I also think it is as much important for the community to learn about the difficulties and challenges, which I believe could be a little bit more specified in the paper. In general, this paper

provides a great set of ideas for the community to benefit from and hopefully it will inspire academic institutions to create evenly innovative and interdisciplinary doctoral programmes.

Specific comments: Paragraph 3.1 'student participation and feedback': Here it is explained that surveys and individual feedback are important in the programme for improvement. It is nice to read that 70% of the students feel happy, however it might be interesting to know why 30% feels less than happy. Is this related to the programme? For example, is it particularly challenging (e.g. in comparison with a traditional PhD, this programme might be more intense/stressful due to a lot of training, meetings, conferences, ..), did the students have different expectations, etc. I also read that cohorts become "happier" and it is considered that the improvements of the programme have something to do with this. Could you give some examples of lessons learnt and such improvements throughout the programme (are they administrative, logistic, technical, social, ..)? What did students particularly indicate in the surveys/feedback that should be improved? I think in the final paragraph 5 'Conclusions and lessons learnt', there could be some more attention to the challenges that were faced during the programme and how they were solved, or how they remain a challenge in this type of programmes. I refer to you the paper "Serlet et al (2020) SMART Research: Toward Interdisciplinary River Science in Europe. Front. Environ. Sci., https://doi.org/10.3389/fenvs.2020.00063" which describes an interdisciplinary doctoral programme (SMART EMJD) that was very successful but also faced many challenges due to a very ambitious set of goals.

The wide interaction of the doctoral students with researchers, industry, practitioners, .. is really a successful key point in this programme and could be more highlighted in the abstract. I really appreciate the out-of-the box type of thinking that was implemented, e.g. the story of the workshop where artists and students collaborated and was then used to integrate the general public.

In line 80 it is written that most students entered the programme after finishing a Master's degree, while some with a Bachelor's degree. Given the age of some students, I would think there are also students who have a working experience? It could be interesting to add this.

Considering the wide range of backgrounds of the students, does this reflect in an equally wide range of thesis topics? In an interdisciplinary programme it can be challenging to have students collaborating e.g. in publications due to a wide range of topics – in contrast to students working in a traditional PhD setting within a team of people working on a single subject.

Did you find specific challenges for integrating the industry on an academic level? I think it is quite 'new' for academic programmes to reach out to the industry on the level of doctoral programmes, particularly for certain disciplines. Are there any recommendations you can give if institutions would like to achieve the same in other countries? Did you make contact yourself, did you advertise the programme, .. ?

I am a bit confused about the number of students, line 75 states 84 students were recruited, line 73 states "all 63 current students", and in the abstract it is said over 70 PhD students will graduate. Perhaps you can also indicate how many students dropped out from 84 recruited?

I did not understand from the paper if the programme is finished, has a limited duration or it will continue indefinitely?

Did the programme have funding to include a living allowance for the students?

Increasingly, the term "doctoral candidate" is being used instead of "doctoral student". This is a little bit related to the culture and local regulations (status). Since the European Commission uses the term doctoral candidate, I would recommend this term.

Technical corrections: Line 95 there is a dash, while in line 96 there is a bracket.

475, 2020.

---

## Author Comment (AC1) · 6 Feb 2021

RESPONSE TO REVIEWER 1

Alyssa Serlet serlet@cerege.fr

GENERAL COMMENTS

COMMENT: I have very much enjoyed reading this paper. I believe doctoral programmes of this nature will be increasing in number in the near future, and papers such as these are of great relevance to the scientific community to learn from the outcomes and experiences of such programmes. The paper is very well written and it explains

thoroughly the need and benefits of the presented programme. It is very important to share the success story of this programme; however, I also think it is as much important for the community to learn about the difficulties and challenges, which I believe could be a little bit more specified in the paper. In general, this paper provides a great set of ideas for the community to benefit from and hopefully it will inspire academic institutions to create evenly innovative and interdisciplinary doctoral programmes.

RESPONSE: Thank you for the positive comments. Adding more discussion on the difficulties and challenges we encountered is indeed a good idea. There are issues like student recruitment and retention that are worth stressing, or the changes we made in response to student comments that we should discuss further. We will add text on these and related issues to the lessons learned section of the paper.

SPECIFIC COMMENTS

COMMENT: Paragraph 3.1 'student participation and feedback': Here it is explained that surveys and individual feedback are important in the programme for improvement. It is nice to read that 70% of the students feel happy, however it might be interesting to know why 30% feels less than happy. Is this related to the programme? For example, is it particularly challenging (e.g. in comparison with a traditional PhD, this programme might be more intense/stressful due to a lot of training, meetings, conferences, ..), did the students have different expectations, etc. I also read that cohorts become "happier" and it is considered that the improvements of the programme have something to do with this.

RESPONSE: In our student survey we found that 27% of students overall assessed themselves as "okay". No student felt "very unhappy" and only 2 students (3%) felt "unhappy", with the reasons for this being known and appropriate support being provided (i. personal reasons; ii. significant project source code errors, subsequently rectified). As stated in our paper, initial data suggests a relationship between stage of PhD programme and general "happiness". We find that each new cohort was happier than the

previous one – hopefully because we improved areas of concern based on student input. We further find that students get significantly more stressed (and less happy) when they come close to their submission time (or end of funding). The WISE CDT invites all students to complete a wide-ranging evaluation survey on completion of the programme, which asks for their feedback on both positive and negative aspects, plus areas for improvement. We will extend the discussion of these results, while accounting for the impact of the pandemic on this evaluation. One issue that is very clear, is that the student very much like the first year of the PhD (with teaching) where they regularly work in a cohort and have a much stronger shared experience than in traditional individual PhD projects.

COMMENT: Could you give some examples of lessons learnt and such improvements throughout the programme (are they administrative, logistic, technical, social, ..)? What did students particularly indicate in the surveys/feedback that should be improved?

RESPONSE: Here are the main actions undertaken in response to student feedback, which we will outline further in the revised manuscript: • Enhanced student support/administrative support; • Seeking Chartered Institution of Water and Environmental Management (CIWEM) accreditation to meet the needs of students without a formal engineering background (they found that this will help their employability with engineering companies); • Amendments to content of 1st year Postgraduate School programme; • Enhancements to transferable skills modules, e.g. viva preparation, careers guidance; • Broadening the Industry Day focus/guests to cover the breadth of students' research interests; • Website enhancements – secure library of CDT templates/guidance; • Ongoing engagement with alumni, including in CDT events, e.g. talks to current students; • Involving students in planning of CDT events.

Student representatives for all cohorts participate in the Open Business section of our regular Programme Management Group meetings to feed in comments and questions from their peers, to propose ideas and to contribute to discussion on planning and programme improvements.

A continuing challenge for students is that the 4 WISE universities' have different regulations / procedures for PhD progression, annual reviews etc. This is unfortunately something we cannot change. To mitigate this issue, we have strengthened our CDT communications with students and have regular partnership meetings, including administrative and finance colleagues.

COMMENT: I think in the final paragraph 5 'Conclusions and lessons learnt', there could be some more attention to the challenges that were faced during the programme and how they were solved, or how they remain a challenge in this type of programmes. I refer to you the paper "Serlet et al (2020) SMART Research: Toward Interdisciplinary River Science in Europe. Front. Environ. Sci., https://doi.org/10.3389/fenvs.2020.00063" which describes an interdisciplinary doctoral programme (SMART EMJD) that was very successful but also faced many challenges due to a very ambitious set of goals.

RESPONSE: Thank you for the suggested reference. We will include the reference and use it as inspiration to extend our discussion of challenges. It looks like our goals (in terms of interdisciplinarity) were not quite as ambitious as in SMART. However, we used the reviewer's comments to review our student guided changes to the program (listed above). We will include and discuss those in the revised manuscript.

COMMENT: The wide interaction of the doctoral students with researchers, industry, practitioners, .. is really a successful key point in this programme and could be more highlighted in the abstract. I really appreciate the out-of-the box type of thinking that was implemented, e.g. the story of the workshop where artists and students collaborated and was then used to integrate the general public.

RESPONSE: Thank you. We will add more on our students' interactions beyond academia, including the art project and industry interactions.

COMMENT: In line 80 it is written that most students entered the programme after finishing a Master's degree, while some with a Bachelor's degree. Given the age of

some students, I would think there are also students who have a working experience? It could be interesting to add this.

RESPONSE: Yes, there are. We will add some information on the work experience of our students.

COMMENT: Considering the wide range of backgrounds of the students, does this reflect in an equally wide range of thesis topics? In an interdisciplinary programme it can be challenging to have students collaborating e.g. in publications due to a wide range of topics – in contrast to students working in a traditional PhD setting within a team of people working on a single subject.

RESPONSE: Students in WISE work on a range of topics. This diversity is likely driven both by their own interests as well as by the diversity of supervisors and their research areas. The connecting tissue (if you like) between the students is the Hydroinformatics aspect of their work. Even though they apply their Hydroinformatics skills to (potentially very) different application areas, they nonetheless regularly share the same computational tools to do so, e.g., optimization and machine learning algorithms or software for uncertainty analysis. For example, students might apply the same sensitivity analysis software to investigate the role of uncertainty in a water treatment model, in a global flood inundation model or in groundwater model. The use of similar tools and methods was one avenue that created connections even for disparate research topics.

COMMENT: Did you find specific challenges for integrating the industry on an academic level? I think it is quite 'new' for academic programmes to reach out to the industry on the level of doctoral programmes, particularly for certain disciplines. Are there any recommendations you can give if institutions would like to achieve the same in other countries? Did you make contact yourself, did you advertise the programme, ..?

RESPONSE: The UK has quite a strong history of academia-industry interactions, including in the context of PhD-level research. Having said this, there is certainly always

a training element in which the academics (and students) have to understand what is considered relevant in the operational world, while industry partners have to understand that even solving their most pressing problem might not always constitute a scientific advancement. Defining a project that is both scientifically novel and operationally relevant is not trivial. One aspect (that we will add to the text) is that the increasing availability of funding to add impact (beyond academia) to a PhD project. Funding sources have been available in recent years to either buy out students for shorter periods (say 3 months) or to add time after the PhD funding has seized so that a better translation of the research work to industry can take place.

COMMENT: I am a bit confused about the number of students, line 75 states 84 students were recruited, line 73 states "all 63 current students", and in the abstract it is said over 70 PhD students will graduate. Perhaps you can also indicate how many students dropped out from 84 recruited?

RESPONSE: We will reconcile and update the numbers in the revised paper.

COMMENT: I did not understand from the paper if the programme is finished, has a limited duration or it will continue indefinitely?

RESPONSE: The student recruitment of new students has finished, but not all students have graduated yet. The latest group of students was recruited in 2018 and is expected to finish in 2022.

COMMENT: Did the programme have funding to include a living allowance for the students? Increasingly, the term "doctoral candidate" is being used instead of "doctoral student". This is a little bit related to the culture and local regulations (status). Since the European Commission uses the term doctoral candidate, I would recommend this term.

RESPONSE: Yes, all students received funding to cover both living allowance and tuition fees. Doctoral candidate is certainly the more generic term given that doctoral

candidates are considered students in some countries and not in others. We will adjust this.

COMMENT: Technical corrections: Line 95 there is a dash, while in line 96 there is a bracket.

RESPONSE: We will correct this.

---

## Referee Comment (RC2) · Anonymous Referee #2 · 19 Mar 2021

General Comments:

Wagener et al. are introducing the doctoral training program WISE running for now 7 years between the Universities of Bath, Bristol, Cardiff and Exeter. They motivate their approach, differing in several well described aspects from the standard 3-year doctoral training, by the need to improve the coding skills of trained water practitioners and scientists to handle the many new data and sensor challenges of the 21st century. They argue that neither a pure water scientist nor a pure computer scientist is fit for the challenges ahead and that their interdisciplinary approach bridges this gap while providing a fruitful and supportive environment for PhD candidates.

The paper is well written, well-structured and easy to read. I fully support sharing outcomes and experiences from new doctoral training approaches to hopefully foster a community learning effect in adapting the often ancient mechanisms of doctoral training.

While I fully appreciate the goal to spread the ideas implemented in the WISE CDT program to potentially inspire other educators, I would wish for the paper to advance from the description of the way the program is working to also offering a bit of general advice and some lessons learned.

Specific Comments:

The last section is named "Conclusions and Lessons Learned". While I can identify the "Why", "What" and outcome I am missing the section on "Lessons Learned". Also, the abstract states that "We conclude with an outlook for PhD training". I would appreciate if the authors could indeed give such an outlook and mention some lessons learned. Please clearly state and condense what in your experience/your surveys identified as helpful and beneficial to the PhD program and should potentially be incorporated in future PhD trainings.

On a similar note - is there a chance to use the ideas of the WISE CDT also for smaller groups/ doctoral programs? What are the "must haves" of modern doctoral training (according to your experience/surveys) that can even be implemented in individual supervising?

The paper currently highlights the positive and well working aspects of the program. But I am sure there must have been several challenges along the way. For the initiators and supervisors as well as for the students. Could you please elaborate on the difficulties and challenges of the process and how they were solved (as already partly described in line 269)?

Regarding the student experience, I was wondering in 3.1. "Student Participation and

Feedback" if there were also less favorable comments in the student surveys. E.g. that students feel lots of pressure, suffer from a high workload or similar? If so, what percentage of the total answers were less favorable? And what was the focus of their concern?

The authors mention different survey results several times. Could you please give a short overview (maybe at the beginning of 3.1.) of the different surveys conducted and specify the scope of those surveys? Is it always the same survey? A yearly ritual? Is it mandatory? What kind of questions are part of the survey? How are the surveys evaluated and used?

Minor/Technical Comments:

It seems only Fig 1 and Fig 6 are referred to in the text. Please add the other figure references to appropriate positions in the text or consider their necessity.

Line 19 – shouldn't this say over 80 candidates as line 375 states 84 students were recruited?

Line 22 – this might be 7 years now?

Line 72 & Line 379 to 406 – are the WISE CDT PhD topics indeed correlating with these ambitious goals? As a reader I would find it helpful to have a short list of example PhD Topics that came out of WISE CDT to assess this myself.

Line 137 – writing 8 as eight?

Line 147 – it would be nice to have a short information in what kind of setting the students were making these comments (probably the/a survey?)

Line 195 – After "Water Hackathron" comes a dash which ends with a bracket

Line 216 – about the "PhD progress monitoring meeting": Who is assessing the progress?

Line 244 – What is meant by regular? Once a month? Once a year? Or at least: What's the aimed for regularity? Same for Line 263.

Line 355 – The authors describe one specific outreach example, but can there be a short introduction to the outreach ambitions of WISE CDT? Is this a specific goal of WISE? Can any other examples at least be named after or before describing the specific example?

Line 377 – I was wondering if the PhD candidates are provided with sufficient funding when entering the program or if this is a separate issue. Do they get stipends? Does funding get better when collaborating with an industry partner?

Line 382 – possibly include reference to Figure 7

---

## Author Comment (AC2) · 29 Mar 2021

RESPONSE TO REVIEWER 2

General Comments:

Wagener et al. are introducing the doctoral training program WISE running for now 7 years between the Universities of Bath, Bristol, Cardiff and Exeter. They motivate their approach, differing in several well described aspects from the standard 3-year doctoral training, by the need to improve the coding skills of trained water practitioners and scientists to handle the many new data and sensor challenges of the 21st century. They

argue that neither a pure water scientist nor a pure computer scientist is fit for the challenges ahead and that their interdisciplinary approach bridges this gap while providing a fruitful and supportive environment for PhD candidates. The paper is well written, well-structured and easy to read. I fully support sharing outcomes and experiences from new doctoral training approaches to hopefully foster a community learning effect in adapting the often ancient mechanisms of doctoral training. While I fully appreciate the goal to spread the ideas implemented in the WISE CDT program to potentially inspire other educators, I would wish for the paper to advance from the description of the way the program is working to also offering a bit of general advice and some lessons learned.

RESPONSE: We thank the reviewer for the positive assessment of our manuscript and for the encouragement to further discuss the wider experiences gained with our WISE CDT. We will be happy to do so and provide some suggestions about how we might do this in our answers below.

Specific Comments:

The last section is named "Conclusions and Lessons Learned". While I can identify the "Why", "What" and outcome I am missing the section on "Lessons Learned". Also, the abstract states that "We conclude with an outlook for PhD training". I would appreciate if the authors could indeed give such an outlook and mention some lessons learned. Please clearly state and condense what in your experience/your surveys identified as helpful and beneficial to the PhD program and should potentially be incorporated in future PhD trainings.

RESPONSE: We agree with the reviewer that we insufficiently discuss our transferrable learning in the current manuscript. We will adjust this section to include a generalization of what we thought worked well and worked less well.

On a similar note - is there a chance to use the ideas of the WISE CDT also for smaller groups/ doctoral programs? What are the "must haves" of modern doctoral training

(according to your experience/surveys) that can even be implemented in individual supervising?

RESPONSE: Yes, there are some ideas and some lessons learned that will also be useful for smaller programs. The ideas of "must haves" is great and we will use it. We, for example, believe that doctoral programs without coordinated training elements (as still the norm in some countries) places students in a disadvantage. This might not be true for the exceptional student, but most students will strongly benefit from wider training.

The paper currently highlights the positive and well working aspects of the program. But I am sure there must have been several challenges along the way. For the initiators and supervisors as well as for the students. Could you please elaborate on the difficulties and challenges of the process and how they were solved (as already partly described in line 269)?

RESPONSE: Yes, the reviewer is absolutely right. Sharing our challenges will be equally helpful to others as will be sharing our positives. We will add main challenges we faced that relate to the program itself. Of course, with over 80 PhD students, there were occasional individual challenges for students or for student-supervisor relations. Such irregular issues were no different from what would be encountered in any reasonably large PhD candidate population. In the revised manuscript we will discuss the following program challenges (and how we delt with them): • Managing a PhD programme as a partnership of 4 universities (given that differences between the universities' approaches are a common "area for improvement" for student requests). • Creating the PG School programme to meet the needs of students with very different academic backgrounds. Students entering the WISE CDT have very heterogeneous backgrounds (engineering, science . . .), which means we regularly had courses where some students struggled (and got stressed) while others were not challenged. This became more difficult after a few years into the program because we had to take out optional modules and make all first-year modules compulsory (due to a change in

teaching structure at the University of Exeter that was out of our control). On the positive side it also meant that students could help each other very well, which added to the cohort feeling. • A key positive element for all students was the cohort experience in year one. Studying as a group rather than as individual PhD candidates was a positive aspect of WISE for everybody. It became difficult to continue this cohort experience once the PhD candidates moved to their home institution at the end of year 1. We tried a range of things to keep this going, which we will discuss. • Managing student visits to international organisations that all have their own administrative and financial requirements created some headaches, though all students went to their chosen destination – until the pandemic started. • Another aspect might be that students not part of WISE might experience less favourable conditions, such as significantly less travel funds. Though there have been no open complaints about this and we tried – when possible – to have other students benefit from WISE activities, e.g. by joining transferrable skills workshops. • Of course, the programme and our students have also inevitably been impacted by the Coronavirus pandemic. Adaptations have been made to ensure the continuation of the training programme and research projects. We will address this in detail in our revision of the manuscript.

Regarding the student experience, I was wondering in 3.1. "Student Participation and Feedback" if there were also less favorable comments in the student surveys. E.g. that students feel lots of pressure, suffer from a high workload or similar? If so, what percentage of the total answers were less favorable? And what was the focus of their concern?

RESPONSE: We will expand on this issue with the following information. Our most comprehensive survey is the Student Experience Survey, which we ask each PhD candidate to complete on conclusion of their programme. This survey asks students to rate their experience of the CDT and seeks their feedback on both positive and negative aspects, plus areas for improvement. We have found that students are able to realistically reflect on their entire CDT experience at that stage (once the final examination stress

is over). To date we have a complete return for Cohort 1 and a partial return for Cohort 2. The survey asks students to rate their overall CDT experience plus diverse elements of the programme using a scale from 1-5 (from "very poor" to "excellent"). Cohort 1 students rated the CDT experience overall as "good", with a mean score of 4.25 out of 5. Cohort 2 respondents to date have scored it 4.5 out of 5. Most frequently mentioned by both cohorts as the best elements are the cohort experience, the funded research visit (up to 3 months abroad), and the opportunity throughout the programme to present work and engage with other researchers. For our complete Cohort 1 return, the most frequently cited areas for improvement were: 1st – Re-think the postgraduate school (which was actioned – we went from 6 compulsory and 2 optional modules to 8 compulsory modules due to University wide changes; we also went from long and thin modules – over a full semester, to short and thick – over a few weeks & we tried to help students who came into the program with less quantitative skills to catch up before the actual program started); 2nd - Unified approach across 4 universities (e.g. registration periods, PhD thesis submission, extension requests) (this is difficult to adjust given that it is largely out of our control); 3rd - More interaction between the 4 universities, both students and academics (e.g. joint supervision, inter-disciplinary events, data/software sharing) (this has developed as WISE grew). The additionally obtained Happiness Index data (limited so far) might suggest lower levels of happiness as students reach the latter stages of their programme and the deadline for PhD thesis submission.

The authors mention different survey results several times. Could you please give a short overview (maybe at the beginning of 3.1.) of the different surveys conducted and specify the scope of those surveys? Is it always the same survey? A yearly ritual? Is it mandatory? What kind of questions are part of the survey? How are the surveys evaluated and used?

RESPONSE: The "Surveys" would cover the following, - End of Year 1: Postgraduate School review meeting – face-to-face feedback to Centre Manager and program Director; - Transferrable Skills and Leadership module evaluation forms; - Annual Progress

Review "Happiness Index"; - End of programme student Experience Survey. Everything is reported to the program management group (directors and co-Is) and discussed there (bi-annually). Data from Happiness Index and Student Experience Surveys are also reported to the external Advisory Board (from Industry (mainly) and Academia). In addition, we informally, we expect student representatives (one per cohort across the four universities) to "survey" their cohort to feed in ideas, comments and criticism to each program management group meeting. Examples of actions taken in response to student feedback: - Enhanced student support/administrative support; - Seeking Chartered Institution of Water and Environmental Management (CIWEM) accreditation to meet the needs of students without a formal engineering background; - Amendments to content of 1st year Postgraduate School programme; - Enhancements to transferable skills modules, e.g. viva (PhD exam) preparation, careers guidance; - Broadening the Industry Day focus/guests to cover the breadth of students' research interests; - Website enhancements – secure library of CDT templates/guidance; - Ongoing engagement with alumni, including in CDT events, e.g. talks to current students; - Involving students in planning of CDT events.

Minor/Technical Comments:

It seems only Fig 1 and Fig 6 are referred to in the text. Please add the other figure references to appropriate positions in the text or consider their necessity.

RESPONSE: That was our oversight. We will refer to them in the text.

Line 19 – shouldn't this say over 80 candidates as line 375 states 84 students were recruited?

RESPONSE: Yes, the numbers were not quite correct. We will revise with the latest numbers.

Line 22 – this might be 7 years now?

RESPONSE: We will be more specific on the time period.

Line 72 & Line 379 to 406 – are the WISE CDT PhD topics indeed correlating with these ambitious goals? As a reader I would find it helpful to have a short list of example PhD Topics that came out of WISE CDT to assess this myself.

RESPONSE: Adding a list of PhD thesis titles is a good idea. We will add this as supplemental material and also organize them in relation to the program goals.

Line 137 – writing 8 as eight?

RESPONSE: OK.

Line 147 – it would be nice to have a short information in what kind of setting the students were making these comments (probably the/a survey?)

RESPONSE: We will add more information on this to the section. Generally, student feedback is obtained later in their programme or on completion of programme through the Student Experience Survey. This particular quote from James Webber was a reflection of how useful the PG School modules turned out to be as his research developed (which he didn't necessarily think at the time of taking them). Similar for Olivia Bailey. "Setting" would be either asking students individually for their feedback and permission to use as quote or through a formal survey.

Line 195 – After "Water Hackathron" comes a dash which ends with a bracket

RESPONSE: We will delete this.

Line 216 – about the "PhD progress monitoring meeting": Who is assessing the progress?

RESPONSE: There are two levels of progress assessment. First, there is the annual assessment taking place at each university separately. Second, there is an annual assessment of candidate progress as part of the program management team, i.e. the director, co-directors and co-Is of WISE.

Line 244 – What is meant by regular? Once a month? Once a year? Or at least:

What's the aimed for regularity? Same for Line 263.

RESPONSE: This will vary somewhat between individual supervisors and time periods within a candidate's PhD. Typical would be every 1-2 weeks.

Line 355 – The authors describe one specific outreach example, but can there be a short introduction to the outreach ambitions of WISE CDT? Is this a specific goal of WISE? Can any other examples at least be named after or before describing the specific example?

RESPONSE: As typical for UK doctoral programs in engineering, there has been a strong focus on interaction with industry throughout the program. Hence most activities beyond academia were focused on strengthening connections and exchange with industry and industry partners. Outreach activities with the general public have been more ad-hoc. We included this specific example because it was particularly successful (based on the PhD candidates' response). Other outreach examples are (we will include examples like these in the revision): • 'Walking with Scientists' – Ioanna Stamataki (Bath Cohort 1) led a guided walking tour showcasing Bath's rich science history – as part of 'FUTURES 2019'. Ioanna's talk focused on the historical floods of the River Avon and the applications of using this data. • New Scientist Live 2019 - WISE and STREAM CDTs joined forces under the umbrella of 'EPSRC Water Engineering' to share their research with the public via posters, demonstrations and experiments • 'Tomorrow's Engineering Week' 2019. As part of this initiative, Cardiff student Santi Lopez (Cohort 5) volunteered on behalf of ICE Wales to provide 'Engineering Team Challenges' to secondary school students, with the aim of encouraging them to consider a career in engineering. • 'TOMORROW': SWINDON'S SCIENCE FESTIVAL. WISE students showcased an Augmented Reality Sandbox, developed by KeckCAVES and supported by the National Science Foundation, which allowed the audience to sculpt miniature sand landscapes and generate 'clouds' and 'rainfall' with their hands. The group also demonstrated the effects of flooding (such as flash flooding from a dam break) and natural disasters (e.g. tsunamis) on different landscapes

and their associated engineering mitigation strategies.

Line 377 – I was wondering if the PhD candidates are provided with sufficient funding when entering the program or if this is a separate issue. Do they get stipends? Does funding get better when collaborating with an industry partner?

RESPONSE: All WISE students are fully with both a stipend and tuition fees for 4 years. All students also have a generous travel budget to be used over the 4-year period for both conference/workshop attendance as well as for an international research visit (∼3 months). Potential additional funds from industrial partners get added as travel or research funds. We will add this information to the manuscript.

Line 382 – possibly include reference to Figure 7

RESPONSE: Yes, we will do that.